# Diet Quality Modulates Gut Microbiota Structure in *Blastocystis*-Colonised Individuals from Two Distinct Cohorts with Contrasting Sociodemographic Profiles

**DOI:** 10.3390/microorganisms13081949

**Published:** 2025-08-21

**Authors:** Claudia Muñoz-Yáñez, Alejandra Méndez-Hernández, Faviel Francisco González-Galarza, Adria Imelda Prieto-Hinojosa, Janeth Oliva Guangorena-Gómez

**Affiliations:** 1Faculty of Health Sciences, Juárez University of the State of Durango, Gómez Palacio 35050, Mexico; claudiamunoz_y@hotmail.com; 2Institute of Genomic Science and Medicine, Torreón 27000, Mexico; alemendez.icm@gmail.com; 3Biomedical Research Center, Autonomus University of Coahuila, Torreón 27000, Mexico; faviel.gonzalez@uadec.edu.mx (F.F.G.-G.); dra.adriaprieto@tec.mx (A.I.P.-H.); 4School of Engineering and Sciences, Tecnologico de Monterrey, Torreón 27000, Mexico

**Keywords:** diet, microbiota, *Blastocystis*, cohorts, 16S rRNA

## Abstract

Diet and gut microbiota are significant determinants of host health, but how dietary quality modulates gut microbiota in *Blastocystis*-colonised individuals remains underexplored. We studied two contrasting cohorts: university students (FACSA, n = 46) and institutionalised children with their caregivers (PAVILA, n = 37), representing distinct dietary and sociodemographic contexts. Eight participants from each cohort tested positive for *Blastocystis*; however, two PAVILA samples could not be sequenced, resulting in a final microbiota subcohort of 14 individuals (FACSA n = 8, PAVILA n = 6). Dietary quality was assessed using the Healthy Eating Index-2020 (HEI-2020), and faecal microbiota was characterised through 16S rRNA sequencing. Alpha and beta diversity were analysed, and genus-level transformed data were further evaluated using permutational multivariate analysis of variance (PERMANOVA), principal coordinates analysis (PCoA), and distance-based redundancy analysis (db-RDA). The FACSA cohort exhibited higher microbial richness and diversity (Shannon and Simpson indexes, *p* < 0.01) compared to PAVILA, with marked differences in microbial composition (PERMANOVA R^2^ = 0.39, *p* = 0.002). Total diet quality correlated with microbial structure (R^2^ = 0.26, *p* = 0.016), with protein (R^2^ = 0.23, *p =* 0.017) and vegetable components (R^2^ = 0.17, *p* = 0.044) as primary contributors. Multivariate analysis showed that higher protein and vegetable intakes were associated with genera such as *Sellimonas*, *Murimonas*, *Alistipes*, and *Desulfovibrio* (FACSA group). In contrast, *Hydrogenoanaerobacterium*, *V9D2013_group*, and *Haemophilus* were linked to lower-quality diets (PAVILA group). Our results indicate that diet quality significantly influences gut microbiota composition in individuals colonised by *Blastocystis*, underscoring its potential as a target for nutritional interventions in vulnerable populations.

## 1. Introduction

The gut microbiota is a dynamic and complex ecosystem that plays a critical role in maintaining human health [1]. It contributes to essential functions such as nutrient metabolism [2], immune regulation [3], and protection against pathogens. Imbalances in this ecosystem (dysbiosis) have been linked to various diseases, including inflammatory bowel disorders, obesity, and metabolic syndrome [4,5]. The composition of the gut microbiota is shaped by multiple intrinsic factors (e.g., age, genetics, and host immunity) and extrinsic factors (e.g., diet, lifestyle, medications, and environment) [6,7,8], with diet considered one of the strongest modulators of microbial diversity and function [9]. Diets rich in fibre, fruits, vegetables, and plant-based foods are consistently associated with higher microbial diversity and enrichment of beneficial taxa, whereas diets high in saturated fats, refined sugars, and ultra-processed foods are linked to reduced microbial richness, inflammation, and metabolic disturbances [7,8]. Beyond diet, host-related factors such as age, lifestyle, and socioeconomic status (SES) also contribute to variations in gut microbial communities [10].

Within this ecosystem, eukaryotic microorganisms also interact with bacterial communities. Among these, the protozoan *Blastocystis* is one of the most common intestinal protists worldwide and has received increasing attention due to the ongoing debate regarding its role in health and disease [11]. Several studies have shown that Blastocystis colonisation can be associated with increased gut bacterial diversity and the enrichment of potentially beneficial taxa such as *Faecalibacterium*, *Alistipes*, and *Prevotella* [12,13]. However, these associations are not consistent across populations, as geographical location, age, dietary habits, and host immunity may modulate the ecological relationship between *Blastocystis* and the gut microbiota. For instance, studies in Colombian children reported no significant differences in microbial composition between colonised and non-colonised individuals, although colonised children exhibited greater microbial richness [14].

Diet is recognised as a major factor shaping gut microbiota structure [9]. High-fibre, plant-based dietary patterns support greater microbial diversity and promote short-chain fatty acid (SCFAs)-producing bacteria such as *Faecalibacterium* and *Roseburia*, which play key roles in maintaining gut integrity and reducing inflammation [15,16]. Intervention studies have demonstrated that the consumption of fibre-rich or fermented foods can modulate the gut microbiota and improve immune function [16,17]. In contrast, Western dietary patterns characterised by low fibre intake and high levels of saturated fats and added sugars have been linked to reduced microbial diversity and an increased prevalence of pro-inflammatory taxa [2]. These dietary patterns may also influence colonisation by intestinal protozoa, including *Blastocystis* [12].

Children living in institutional settings such as orphanages, group homes, or low-income households often depend on donated or program-provided food, which are factors that affect both diet quality and gut microbial ecology. Systematic reviews have shown that nutritional intake in these contexts is frequently inadequate and lacks diversity, predisposing children to malnutrition and limited dietary quality [18,19]. Limited access to diverse foods and healthcare services, combined with repeated exposure to enteric pathogens, can impair microbiota development and resilience [20,21]. Socioeconomic status has also been shown to influence microbial profiles, with children from low-SES environments displaying a higher prevalence of taxa such as *Prevotella* and *Escherichia-Shigella*, while children from high-SES settings often have a greater abundance of *Bifidobacterium* and *Lactobacillus* [22]. Malnutrition and parasitic infections are particularly common in resource-limited contexts, contributing to chronic inflammation, nutrient malabsorption, and stunted growth [23,24]. In this context, *Blastocystis* colonisation may reflect environmental and nutritional exposures rather than pathogenicity. While the relationship between *Blastocystis* and the microbiota has been partially described, the role of dietary quality in modulating this interaction remains poorly understood. We hypothesised that the impact of *Blastocystis* on gut microbiota composition and diversity would vary according to diet quality and sociodemographic environment. Therefore, this study aimed to examine whether gut microbiota structure differs according to diet quality in *Blastocystis*-colonised individuals from two cohorts with contrasting dietary patterns and sociodemographic contexts: university students (FACSA) and institutionalised children with their caregivers (PAVILA).

## 2. Materials and Methods

### 2.1. Study Population and Design

This cross-sectional study analysed two cohorts. The PAVILA cohort comprised 37 participants—27 children living in a group home and 10 adult caregivers—resulting in a child-to-caregiver ratio of 2.7:1. Among them, six children and two caregivers tested positive for *Blastocystis*. Although this specific information is not presented in Tables 1 and 2 or the Appendix A, it was obtained from the cohort’s raw data. The FACSA cohort included 46 university students (32 females and 14 males), with a median age of 21 years (Q1: 20, Q3: 22). In the PAVILA cohort, 22 participants were female and 15 were male, with a median age of 13 years (Q1: 7, Q3: 27). Complete demographic information, including age distribution, gender, and anthropometric data, is provided in Appendix A. Participants were recruited voluntarily and provided informed consent in accordance with institutional ethical standards. Demographic information, including age, sex, and anthropometric measurements, was collected. Diet quality was assessed using a validated food frequency questionnaire.

For the gut microbiota analysis, a subset of 14 individuals colonised by *Blastocystis* was included: eight from the FACSA cohort and six from the PAVILA cohort. Two additional samples from PAVILA were excluded due to poor DNA integrity, which impeded successful sequencing. This exploratory analysis was designed to assess whether diet quality influences gut microbiota composition across contrasting host environments in individuals colonised by Blastocystis.

### 2.2. Dietary Assessment

A validated Food Frequency Questionnaire (FFQ) developed by Macedo et al. in 2013 was used [25] to evaluate diet quality. Nutritionists administered the FFQ to participants via Google Docs. Reported food group portions were converted into grams per day and analysed for energy and macronutrients using the Evalfinut 2.0 software (Ibero-American Nutrition Foundation, FINUT). This program assesses individual dietary intake based on food frequency and nutrient composition according to the Spanish Food Composition Database (BEDCA 2.0) [26] and the USDA National Nutrient Database for Standard Reference (Version 28) [27].

The Healthy Eating Index (HEI-2020) was calculated for each participant to assess their diet quality [28]. The International Physical Activity Questionnaire (IPAQ) was used to evaluate Physical activity, classifying individuals into low, moderate, or high activity levels based on self-reported time spent in vigorous and moderate activity, walking, and sedentary time. Physical activity levels were expressed in Metabolic Equivalent of Task (MET) minutes per week, in accordance with standard IPAQ scoring protocols [29].

### 2.3. Sample Collection and DNA Extraction

Stool samples were collected from participants in sterile 2 mL vials and stored at −80 °C until processing. Total bacterial genomic DNA was extracted from 200 mg of stool using the E.Z.N.A.^®^ Stool DNA Kit (OMEGA Bio-Tek, Inc., Norcross, GA, USA), following the manufacturer’s instructions. DNA integrity was assessed by 1.5% agarose gel electrophoresis, and DNA concentration was measured using a NanoDrop (Thermo Fisher Scientific Inc., Waltham, MA, USA) spectrophotometer.

### 2.4. Gut Microbiota Profiling

Microbiota profiles were analysed in fourteen faecal DNA samples from *Blastocystis*-colonised subjects: eight from the FACSA cohort and six from the PAVILA cohort. Two samples of the PAVILA cohort were excluded due to low DNA quality. For each sample, the V3 hypervariable region of the 16S rRNA gene was amplified using primers containing Illumina overlap adapters (forward: 5′-TCGTCGGCAGCGTCAGATGTGTATAAGAGACAGACYCCTACGGGRGGCAGCAG-3′ and reverse: 5′-GTCTCGTGGGCTCGGAGATGTGTATAAGAGACAGTTACCGCGGCTGCTGGCAC-3′). Amplicons were purified with AMPure XP beads and subsequently indexed using the Nextera XT v2 kit to add dual indices. DNA concentrations were quantified using a Qubit 3.0 fluorometer (Thermo Fisher Scientific Inc., Waltham, MA, USA). Libraries were then normalised and pooled in equimolar amounts, followed by sequencing on the Illumina MiSeq^TM^ platform using the MiSeq^®^ Reagent Kit V2 (Illumina, Inc., San Diego, CA, USA) (500 cycles) to generate 250 bp paired-end reads [30].

### 2.5. Bioinformatic Processing

Initial quality control of raw reads was performed using Fast Quality Control (FastQC). Forward and reverse reads were trimmed to 240 base pairs based on quality scores. The trimmed reads were then merged and denoised using the Divisive Amplicon Denoising Algorithm (DADA2) plugin in Quantitative Insights Into Microbial Ecology 2 (QIIME2), enabling the generation of high-resolution amplicon sequence variants (ASVs). Taxonomic assignment was conducted using a closed-reference approach with the SILVA rRNA gene database project reference (99% similarity threshold). Chimeric sequences were removed using the Vectorised Search Tool (VSEARCH) plugin. Taxonomic classification was carried out with a Naive Bayes classifier trained on SILVA sequences. Phylogenetic reconstruction involved multiple sequence alignment with Multiple Alignment using Fast Fourier Transform (MAFFT) and tree construction using FastTree (approximately-maximum-likelihood phylogenetic trees), version 2.1.11 through the q2-phylogeny module. Alpha diversity was assessed using the Shannon and Simpson indices, while beta diversity was calculated using Bray–Curtis distances.

### 2.6. Detection of Blastocystis

The presence of *Blastocystis* DNA from faecal DNA extracts was determined using a real-time Polymerase Chain Reaction (PCR) assay with a specific probe. The primers used were F1: GGTCCGGTGAACACTTTGGATTT and R1: CCTACGGAAACCTTGTTAC-GACTTCA, with the probe FAM-TCGTGTAAATCTTACCATTTAGAGGA-MGBNFQ (Integrated DNA Technologies, IDT). Reactions were performed in a RotorGene^®^ thermal cycler (Qiagen, Hilden, Germany) using the QuantiNova Probe RT-PCR (Qiagen, Hilden, Germany) kit in 10 µL volumes containing 5 µL of 2× master mix, 0.1 µL of RT mix, 0.5 µL of primer and probe mix, 3.5 µL of RNase/DNase-free water, and 1 µL of DNA (40 ng). Thermal cycling conditions followed the manufacturer’s protocol: 95 °C for 5 min, followed by 40 cycles of 95 °C for 5 s and 60 °C for 30 s [31]. The assay was performed as a qualitative test to determine the presence or absence of *Blastocystis*. No assessment of colonisation intensity was performed, as the analysis of parasite load was beyond the scope of this study.

### 2.7. Statistical Analysis

Dietary characteristics between the FACSA and PAVILA cohorts were compared using the HEI-2020 Healthy Eating Index, considering both the total score and the scores of its 13 individual components. Quantitative variables were summarised as means ± standard deviations or medians with interquartile ranges, depending on the distribution of the data. Normality was assessed using the Shapiro–Wilk test. Based on data distribution, comparisons between cohorts were conducted using either the Student’s t-test or the Mann–Whitney U test. Statistical significance was set at *p* < 0.05, and *p*-values were adjusted for multiple comparisons using the Benjamini–Hochberg method. For microbiota data, alpha diversity analysis was conducted on genus-level abundance data transformed using the Hellinger transformation. The Shannon and Simpson indices were calculated to evaluate community richness and evenness. Beta diversity was assessed using Bray–Curtis distances, also based on Hellinger-transformed data. PERMANOVA (Permutational Multivariate Analysis of Variance) was used to test whether overall gut microbiota composition differed significantly according to the total HEI-2020 score and its individual components. Additionally, distance-based Redundancy Analysis (db-RDA) was performed to visualise associations between dietary variables and microbial community variation, with particular emphasis on key components such as vegetable and protein intake. The ten taxa most strongly associated with the ordination axes (*p* < 0.1) were highlighted graphically. All statistical and graphical analyses were performed using R (version 4.4.3).

### 2.8. Ethical Considerations

The Research Ethics Committee of the Faculty of Medicine and Nutrition at Juárez University of Durango, under registration number CEI-FAMEN-36, approved the study. All procedures were carried out according to the ethical principles outlined in the Declaration of Helsinki and in compliance with current national regulations governing research involving human subjects.

Adult participants provided written informed consent prior to inclusion in the study. The legal representatives of the children in the PAVILA cohort signed the consent letters, as well as assent from minors, considering their age and level of understanding. The confidentiality of personal data was ensured through the code and the secure storage of biological samples. Participation was voluntary and posed no physical or psychological risks to the subjects.

## 3. Results

Given the limited sample size (n = 14), the following results should be interpreted as exploratory and hypothesis-generating, rather than conclusive.

### 3.1. Dietary Quality and Blastocystis Status by Cohort

We first compared HEI-2020 dietary component scores between individuals with *Blastocystis*-colonisation within each cohort. In the FACSA cohort, colonised participants exhibited higher total HEI-2020 scores, with a tendency toward greater intake of fruits and vegetables (Table 1). In contrast, in the PAVILA cohort, colonised children showed significantly lower vegetable intake compared to non-colonised peers (adjusted *p* = 0.018, Table 2).

**Table 1 microorganisms-13-01949-t001:** HEI-2020 component scores by *Blastocystis* status in the FACSA cohort.

Component	*Blastocystis* Present n = 8	*Blastocystis* Absent = 38	*p*-Value	Adjusted *p*-Value
Energy intake	2812.75 ± 721.59	2728.89 ± 950.18	0.522	0.642
Caloric activity	2490.50 ± 622.20	2449.11 ± 650.75	0.805	0.882
Fruits	4.01 ± 1.21	2.82 ± 1.65	0.071	0.298
Whole fruits	1.56 ± 0.96	1.38 ± 1.74	0.195	0.390
Vegetables	2.53 ± 1.25	1.81 ± 0.77	0.172	0.390
Legumes	2.83 ± 0.79	3.35 ± 1.42	0.184	0.390
Whole grains	9.03 ± 2.43	7.75 ± 3.41	0.331	0.530
Dairy	4.12 ± 2.39	3.78 ± 2.47	0.503	0.642
Protein foods	3.32 ± 0.41	3.21 ± 0.53	0.485	0.642
Seafood/plant protein	3.28 ± 1.72	3.21 ± 1.64	0.835	0.882
Fatty acid ratio	4.77 ± 1.43	3.16 ± 2.20	0.020 *	0.163
Refined grains	9.18 ± 1.25	8.08 ± 2.32	0.262	0.466
Added sugars	1.10 ± 2.03	2.40 ± 2.58	0.193	0.390
Saturated fat	8.33 ± 2.12	6.15 ± 3.03	0.075	0.298
Total HEI-2020	64.03 ± 2.36	56.67 ± 9.91	0.005 *	0.079

HEI-2020 component scores (mean ± SD) by *Blastocystis* status in the FACSA cohort. No statistically significant differences were found after *p*-value adjustment. Adjusted *p*-value corrected by the Benjamini-Hochberg method. Significance: * ≤ 0.05, ns > 0.05.

**Table 2 microorganisms-13-01949-t002:** HEI-2020 component scores by *Blastocystis* status in the PAVILA cohort.

Component	*Blastocystis* Present n = 8	*Blastocystis* Absent n = 29	*p*-Value	Adjusted *p*-Value
Energy intake	4678.75 ± 2245.32	3175.45 ± 1459.14	0.140	0.681
Caloric activity	2070.25 ± 396.57	2148.70 ± 752.25	0.981	0.981
Fruits	2.67 ± 2.06	3.50 ± 1.45	0.225	0.681
Whole fruits	2.29 ± 2.06	2.30 ± 1.77	0.961	0.981
**Vegetables**	**0.55 ± 0.49**	**1.70 ± 0.72**	**0.001 ****	**0.018 ***
Legumes	2.65 ± 1.98	3.40 ± 1.68	0.426	0.681
Whole grains	7.86 ± 3.51	6.41 ± 3.71	0.468	0.681
Dairy	4.57 ± 1.79	4.66 ± 3.05	0.788	0.901
Protein foods	2.72 ± 0.66	2.91 ± 0.63	0.654	0.805
Seafood/plant protein	2.62 ± 0.51	1.72 ± 1.82	0.246	0.681
Fatty acid ratio	4.58 ± 3.41	3.43 ± 3.35	0.445	0.681
Refined grains	7.78 ± 3.41	6.48 ± 3.21	0.412	0.681
Added sugars	2.79 ± 2.74	1.37 ± 3.24	0.065	0.520
Saturated fat	6.69 ± 4.80	6.09 ± 3.76	0.607	0.805
Total HEI-2020	57.13 ± 7.20	53.78 ± 11.78	0.434	0.681

HEI-2020 component scores (mean ± SD) by *Blastocystis* status in the PAVILA cohort. A significant difference was observed in the ‘Vegetables’ component after *p*-value adjustment. Adjusted *p* value: corrected by the Benjamini-Hochberg method. Significance codes: ** ≤ 0.01, * ≤ 0.05, ns > 0.05.

### 3.2. Alpha and Beta Diversity Analysis Between FACSA and PAVILA Cohorts

To characterise the gut microbiota structure in participants from the two cohorts, FACSA (university students) and PAVILA (children and caregivers in a shelter), we evaluated three alpha diversity metrics: observed richness (number of genera), Shannon diversity index, and Simpson diversity index.

Our results showed that the FACSA cohort exhibited significantly greater richness and diversity than the PAVILA cohort. Specifically, the observed number was higher in the FACSA group (*p* = 0.0024), as was the Shannon index (*p* = 0.0013), suggesting a more diverse and evenly distributed gut microbial community. The Simpson index was also significantly different between cohorts (*p* = 0.0013), further supporting the presence of compositional differences in microbial communities (Figure 1).

Figure 2 illustrates a clear separation in microbiota composition between the FACSA and PAVILA cohorts along the first two principal coordinates. Each point represents an individual sample, while shaded ellipses denote the 95% confidence interval for each group. The first two principal coordinates analysis (PCoA) axes explained 50.55% and 14.29% of the total variance, respectively. The PERMANOVA test confirmed a significant difference between groups (F = 7.15, R^2^ = 0.374, *p* = 0.001), supporting the hypothesis that host-related factors contribute to distinct gut microbial community structures.

### 3.3. PCoA Biplot with Genus-Level Arrows

The PCoA based on Bray–Curtis dissimilarities and Hellinger-transformed genus-level data revealed a clear separation between the two cohorts, despite all individuals being colonised by *Blastocystis* (Figure 3). Samples from the FACSA cohort, with higher HEI-2020 scores and healthier dietary patterns, clustered predominantly toward the right side of the ordination space. In contrast, individuals from the PAVILA cohort, who had lower overall dietary quality, grouped toward the left. Genus-level vectors overlaid on the biplot illustrate the taxa most strongly associated with this compositional variation. Notably, most genera including *Bacteroides*, *Odoribacter*, *Subdoligranulum*, *Murimonas*, *Faecalibacterium*, and *Parabacteroides* were projected toward the FACSA cluster, suggesting an enrichment of these taxa in individuals with better diet quality. Surprisingly, only *Segatella* was directionally associated with the PAVILA cluster (Figure 3). This pattern suggests that the microbiota of university students (FACSA), despite colonisation by *Blastocystis*, remains more diverse and enriched in genera typically linked to metabolic health and dietary fibre intake.

These results underscore the modulatory role of diet quality in shaping gut microbial communities, even among *Blastocystis*-colonised hosts, and suggest the ecological impact of contrasting host environments such as institutionalisation and youth in the PAVILA cohort versus independent adulthood in the FACSA cohort. Moreover, when referring to the “ecological impact” of host environments, we refer to the combined influence of factors such as institutionalisation, age, dietary control, and environmental exposure, which jointly shape the gut microbiota landscape in each cohort.

### 3.4. Diet–Microbiota Associations Within Colonised Individuals

#### 3.4.1. Beta Diversity and Diet Quality Score in Blastocystis-Colonised Individuals

To evaluate whether diet quality influences overall gut microbial composition in individuals colonised by Blastocystis, we performed a PERMANOVA using Bray–Curtis dissimilarities. We modelled the total HEI-2020 diet quality score as a continuous predictor variable. The PERMANOVA test revealed a statistically significant association between diet quality and microbiota structure (R^2^ = 0.24, F = 3.86, *p* = 0.011), indicating that dietary quality accounts for approximately 24% of the variance in microbial community composition. Figure 4 illustrates a visible compositional gradient, where individuals with lower diet scores tend to cluster separately from those with higher scores. The first two axes explain 50.55% and 14.29% of the total variance, respectively.

#### 3.4.2. Microbial Composition Projected onto Dietary Quality Gradient

To analyse the relationship between diet quality and gut microbiota composition, PCoA based on Bray–Curtis dissimilarities was performed and coloured by HEI-2020 scores. The first two axes explained 50.55% and 14.29% of the total variance, respectively. A compositional gradient was observed: individuals with lower HEI-2020 scores (blue hues) were positioned mainly on the left side of the ordination space, whereas those with higher scores (red hues) clustered toward the right, both in the upper and lower quadrants.

An envfit projection was used to identify the top 10 genera most associated with this gradient (*p* < 0.1). *Sellimonas* and *Subdoligranulum* were strongly associated with higher HEI-2020 scores. At the same time, *Parabacteroides*, *Odoribacter*, *Ocillospira*, *Bacteroides*, and *Parasuterella* also pointed in that direction, suggesting a shared ecological niche among participants with higher-quality diets. In contrast, *Segatella* was the only genus whose vector aligned with the cluster of individuals with lower HEI-2020 scores, suggesting a potential association with poorer diet quality (Figure 5).

#### 3.4.3. PERMANOVA Results by Dietary Component

We performed PERMANOVA using Bray-Curtis dissimilarities calculated from Hellinger-transformed genus-level microbiota data to assess the influence of each dietary component on microbial community structure. As shown in Table 3, significant associations were observed for the protein intake score (R^2^ = 0.232, *p* = 0.017) and the vegetable intake score (R^2^ = 0.167, *p* = 0.044). These findings indicate that these two components of diet quality may be key contributors to the observed variation in gut microbiota composition. The remaining components showed weaker and non-significant relationships with microbial structure.

Among HEI-2020 components, the protein foods score and total vegetables score were associated with microbial community structure (PERMANOVA: protein foods, R^2^ = 0.232, F = 3.63, *p* = 0.017; total vegetables, R^2^ = 0.167, F = 2.40, *p* = 0.044; Table 3) in individuals colonised by *Blastocystis*.

Table 3 PERMANOVA results evaluate the association between individual dietary components from the Healthy Eating Index-2020 (HEI-2020) and gut microbiota composition based on Bray-Curtis dissimilarities of Hellinger-transformed genus-level data. Statistically significant associations (*p* < 0.05) were observed between the protein foods score and the total vegetable score, indicating their influence on microbial community structure among individuals colonised by *Blastocystis*. R^2^ represents the proportion of variance explained by each dietary variable.

#### 3.4.4. db-RDA Analysis: Protein and Vegetable Intake

We conducted distance-based redundancy analysis (db-RDA) using Bray-Curtis dissimilarities of Hellinger-transformed genus-level microbiota data constrained by specific HEI-2020 dietary components.

To analyse the relationship between protein intake and gut microbiota composition in *Blastocystis*-colonised individuals, a distance-based redundancy analysis (db-RDA) was performed using Bray–Curtis dissimilarities of Hellinger-transformed genus-level data, constrained by the protein foods component of the Healthy Eating Index-2020 (HEI-2020). The first canonical axis (CAP1) explained 100% of the constrained variance, while the first unconstrained axis (MDS1) accounted for 37.06% of the residual variance.

A clear cohort-based separation was observed along CAP1: individuals from the FACSA cohort (orange), characterised by higher protein scores, clustered to the right, while those from the PAVILA cohort (green), characterised by lower protein intake, grouped on the left. Environmental vector fitting (envfit, *p* < 0.1) identified ten genera strongly associated with the ordination axes. Genera such as *Sellimonas*, *Subdoligranulum*, *Bacteroides*, and *Murimonas* were aligned with higher protein intake and the FACSA cohort. Conversely, *Segatella* was the only genus that projected in the direction of lower protein scores, aligning with the PAVILA group. (Figure 6). These findings suggest a diet-driven structuring of the gut microbiota that reflects differences in protein consumption between cohorts.

To explore the influence of vegetable intake on gut microbiota composition in *Blastocystis*-colonised individuals, a distance-based redundancy analysis (db-RDA) was conducted using Bray–Curtis dissimilarities of Hellinger-transformed genus-level data, constrained by the vegetable component score of the Healthy Eating Index-2020 (HEI-2020). The first constrained axis (CAP1) explained 100% of the variance associated with vegetable intake, while the first unconstrained axis (MDS1) accounted for 37.06% of the residual variance.

A clear separation between cohorts was observed: individuals from the FACSA cohort (orange), characterised by higher vegetable scores, were positioned on the right side of CAP1, whereas those from the PAVILA cohort (green), with lower scores, clustered on the left. The environmental fit analysis (envfit, *p* < 0.1) identified several genera associated with the CAP1 gradient. Taxa such as *Sellimonas*, *Acutalibacter*, *Murimonas*, and *Alistipes* were aligned with higher vegetable intake in the FACSA cohort, while *Segatella* was projected in the direction of lower vegetable intake in the PAVILA group. Figure 7. These findings suggest a compositional shift in the microbiota structure linked to dietary vegetable intake among *Blastocystis*-positive individuals.

#### 3.4.5. Multivariate Analysis of the Combined Effect of Protein and Vegetable Intake on Gut Microbiota Composition in Individuals Colonised by Blastocystis

A multivariate model incorporating protein and vegetable dietary scores demonstrated a significant impact on gut microbiota composition (R^2^ = 0.31, F = 2.50, *p* = 0.025). This result suggests that, in *Blastocystis*-colonised individuals, specific dietary components collectively contribute to explaining more than 30% of the variability in microbial structure, reinforcing their role as key ecological determinants. Table 4.

We observed a clear separation between the FACSA and PAVILA cohorts in the db-RDA analysis along the CAP1 axis based on the combined protein and vegetable dietary score. The model was statistically significant (*p* = 0.025), indicating that these variables explain an important proportion of the variation in the gut microbiota of individuals colonised by *Blastocystis*. The projected vectors showed that genera such as *Sellimonas*, *Murimonas*, *Alistipes*, and *Desulfovibrio* were associated with better dietary scores and oriented toward the FACSA cohort. In contrast, genera such as *Hydrogenoanaerobacterium, V9D2013_group*, and *Haemophilus* aligned with individuals from the PAVILA cohort, who present lower vegetable and protein intakes. This pattern supports that diet exerts a structuring effect on the microbiota, modulating the microbial community even in the common presence of *Blastocystis* (Figure 8).

It is worth noting that all individuals included in this analysis were colonised by *Blastocystis*, yet their gut microbial composition varied markedly depending on diet quality and cohort. This may suggest that *Blastocystis* colonisation alone does not define a consistent microbiota pattern, and that dietary context may shape distinct microbial signatures even within colonised populations. These findings reinforce the notion that the relationship between *Blastocystis* and gut microbiota is not uniform and may be modulated by host-related factors such as diet, age, and living conditions.

## 4. Discussion

In this study, we analysed gut microbiota composition exclusively in individuals colonised by *Blastocystis*, drawn from two contrasting cohorts: university students (FACSA) and institutionalised children and caregivers (PAVILA). Although colonisation was present in both cohorts, it occurred independently of vegetable intake or overall diet quality, suggesting that other environmental or host-related factors may influence colonisation.

### 4.1. Diet Quality and Cohort Differences

Diet quality, as quantified by HEI-2020 scores, differed significantly between the two cohorts, with FACSA participants demonstrating higher overall diet quality. FACSA participants consumed higher amounts of fruits, healthier fats (as reflected in favourable ratios of unsaturated to saturated fatty acids), and lower levels of added sugar patterns known to support gut microbiota diversity and metabolic health [32]. In contrast, PAVILA children exhibited lower vegetable intake, which aligns with studies in diverse populations showing that insufficient consumption of vegetables is associated with reduced microbial richness and functional metabolites like SCFAs [33]. These observations support the notion that *Blastocystis* colonisation can occur across diverse dietary backgrounds. However, the distinct microbial signatures observed in each cohort suggest that diet quality and host context, rather than colonisation status alone, are more strongly associated with microbial diversity. This aligns with findings from studies in multi-ethnic populations, where HEI components, particularly fruit and vegetable intake, account for a substantial variance in gut microbial community structure [7,33,34]. Interestingly, the relationship between diet quality and *Blastocystis* colonisation varied between cohorts. In FACSA, colonised individuals had higher HEI-2020 scores, consistent with previous studies suggesting that diets rich in fibre and vegetables may promote colonisation [35]. Conversely, in PAVILA, colonised children showed lower vegetable intake, indicating that other factors such as hygiene, immune development, or environmental exposures may be more influential in determining colonisation in vulnerable, institutionalised settings. These findings align with epidemiological studies showing that *Blastocystis* prevalence is particularly high in settings characterised by poor sanitation, low socioeconomic status, and limited education. For example, research from Peru reported a strong link between the use of latrines instead of flush toilets and higher infection rates [36]. Another study demonstrated significant associations between *Blastocystis* infection and low socioeconomic and educational background, as well as poor hygiene conditions [37]. This contrast highlights that the link between diet and *Blastocystis* is likely context-dependent and modulated by broader sociodemographic conditions.

### 4.2. Alpha Diversity Insights

When comparing gut microbiota diversity among *Blastocystis*-colonised individuals, we observed that participants from the FACSA cohort exhibited significantly higher microbial richness and evenness than those from the PAVILA cohort, suggesting a strong influence of host-related factors. While previous studies have reported that *Blastocystis* colonisation is associated with increased microbial diversity in healthy hosts [38,39], this association may not be universal. For instance, Tito et al. (2019) reported that colonisation was positively correlated with greater microbial richness and enrichment in *Clostridiales* and *Ruminococcaceae* taxa [40]. Similarly, Audebert et al. (2016) found that *Blastocystis*-positive individuals had a higher abundance of *Prevotella*- or *Ruminococcus*-driven enterotypes and greater alpha diversity than negative subjects [38]. However, in our study, dietary quality and host environment appeared to play a more prominent role in shaping microbial diversity, even in the presence of *Blastocystis*. These observations support the view that colonisation can occur across various dietary contexts. However, broader lifestyle and nutritional factors ultimately modulate the overall microbial profile.

Furthermore, it is well established that gut microbiota alpha diversity increases substantially during early childhood and continues to mature through early school age. A longitudinal study by Stewart et al. demonstrated that bacterial taxonomic diversity increases significantly from infancy to approximately 36 months, stabilizing thereafter [41]. Similar findings were reported by Roswall et al., indicating gradual diversity gains during the first decade of life [42].

These patterns suggest that the lower alpha diversity observed in the younger, institutionalised PAVILA cohort can be attributed to the ongoing maturation of the microbiome, a process delayed compared to adult levels, rather than external speculation. The evidence underscores that age and colonisation status together influence microbial ecological outcomes.

### 4.3. Beta Diversity and Ecological Context

When analyses were confined to *Blastocystis*-colonised individuals, we observed significant differences in community composition between cohorts (PERMANOVA *p* = 0.001; Figure 2). This suggests that factors such as lifestyle and institutional living exert independent effects on gut microbiota structure. Longitudinal cohort studies demonstrate that beta diversity evolves across the lifespan, with adult-like microbiota profiles typically emerging by the end of the first decade of life [42]. Additionally, lifestyle factors such as socioeconomic conditions and dietary diversity are well documented to influence microbial composition substantially, independent of age; institutional environments and constrained dietary options have also been associated with reduced microbial diversity and shifts in community composition in preschool-age children [32,43,44,45,46].

### 4.4. Exploratory Visualisation of Diet–Microbiota Interactions in Blastocystis-Colonised Individuals

A principal coordinates analysis (PCoA) based on Bray–Curtis dissimilarities and coloured by HEI-2020 total scores revealed a structured distribution of samples, despite all individuals being colonised by *Blastocystis*. Notably, individuals with higher dietary quality (as indicated by red hues) clustered toward one side of the ordination space, while those with lower scores (blue/purple hues) were positioned on the opposite end (as shown in Figure 5). This gradient-like pattern suggests that even within a uniformly colonised population, dietary quality may drive compositional shifts in gut microbiota. These findings suggest that *Blastocystis* colonisation can occur in diverse dietary contexts and that host diet plays a modulatory role in shaping microbial ecology. It should be noted that the visualisation of microbiota composition constrained by HEI-2020 scores revealed taxa-specific associations along the dietary quality gradient. Notably, *Segatella* was enriched in individuals with low-quality diets, which contrasts with the previously reported role in SCFAs production and metabolic health [47,48]. The positioning of *Segatella* along the HEI-2020 gradient revealed a strong association with individuals exhibiting the lowest overall dietary quality. Interestingly, this genus was also detected in the PAVILA cohort, where colonised individuals had notably lower vegetable intake. This apparent paradox suggests that *Segatella* may respond to broader ecological conditions influenced by multiple dietary components, rather than isolated vegetable consumption. Indeed, *Segatella* isolates have been shown to ferment plant polysaccharides into diverse short-chain fatty acids (formate, lactate, succinate, propionate, acetate) [49], indicating sensitivity to complex dietary substrates. This finding is consistent with reports suggesting that *Segatella* may increase in environments where diets are high in sugars and low in protein, which are patterns commonly observed in resource-limited settings. As noted by Xiao et al. (2024), *Segatella* dominance does not necessarily indicate good nutritional quality; rather, it may reflect a monotonous diet rich in plant-based carbohydrates but lacking in variety and essential nutrients [50]. The coexistence of *Segatella* with *Blastocystis* in the PAVILA cohort may suggest a standard ecological configuration associated with institutional diets that have high starch content but low nutritional diversity. This highlights the importance of understanding microbial profiles within the specific social and dietary contexts of each group.

On the other hand, principal coordinates analysis (PCoA) also revealed a transparent gradient, where individuals with higher-quality diets occupied a different region in the ordination space. Projecting microbial genera using envfit analysis further supported this pattern: genera such as *Sellimonas* and *Subdoligranulum* were enriched in participants with higher HEI-2020 scores, consistent with their potential role in beneficial ecological setups. Interestingly, other taxa, such as *Acutalibacter* and *Bacteroides*, also aligned with this higher-quality dietary group. In a systematic review of dietary interventions in individuals with type 2 diabetes, Bock et al. (2022) reported an increase in the abundance of Bacteroides and *Alistipes* following intervention with Mediterranean or high-fibre diets. *Subdoligranulum* exhibited variable responses, depending on the type of intervention and the basal microbiota; these findings partially align with our results. In our cohort of healthy individuals colonised by *Blastocystis*, we also observed that Bacteroides and *Alistipes* align with patterns of higher dietary quality. Although the metabolic context differs, these observations suggest that specific genera may serve as common functional indicators in response to dietary improvements [51]. A previous study analysing colonic mucosal samples found that participants with lower HEI scores had significantly reduced abundances of *Subdoligranulum* and *Parabacteroides* compared to those with better diet quality. Furthermore, they observed that *Alistipes* tended to be more abundant in individuals with greater adherence to healthy dietary patterns, characterised by higher consumption of fruits, fibre, and greater dietary diversity [52]. Although Blastocystis colonised both cohorts, the observed differences in microbial composition appear more strongly associated with diet quality and host environment. In particular, the FACSA cohort showed a richer and more diverse microbiota, enriched in genera associated with fibre metabolism and metabolic health. This does not rule out a role for *Blastocystis*, but suggests that its impact may be modulated or even masked by other dominant factors such as dietary intake and host lifestyle.

### 4.5. Dietary Components Shape Microbiota

The db-RDA constrained by the HEI-2020 protein component suggests that protein intake modulates gut microbiota composition in individuals colonised by *Blastocystis*. The observed separation of cohorts along CAP1 is consistent with dietary records showing higher protein food scores in the FACSA group relative to PAVILA. Notably, the genera *Sellimonas*, *Subdoligranulum*, and *Bacteroides*, which projected toward the FACSA cluster, have been previously associated with greater dietary diversity and fibre-protein co-fermentation (ref).

Interestingly, *Segatella* was again aligned with the PAVILA group and lower protein intake, consistent with its prior association with lower vegetable scores in this same population. This recurring pattern suggests that *Segatella* may represent a microbial marker of low-nutrient dietary profiles [50]. Together, these findings highlight the relevance of individual dietary components beyond total diet quality in shaping gut microbial ecology under *Blastocystis* colonisation, potentially modulating colonisation dynamics or host-microbe interactions.

PERMANOVA results revealed that protein intake (R^2^ = 0.23, *p* = 0.017) and vegetable intake (R^2^ = 0.17, *p* = 0.044) were significantly associated with microbiota composition. This aligns with evidence that dietary protein strongly influences gut microbial community structure and metabolic outputs: a high-protein diet can modulate bacterial taxa involved in proteolytic fermentation, affecting SCFAs and potentially driving changes in gut health [53]. Plant-based components, including vegetables rich in fibre, have been shown to promote the growth of SCFAs-producing bacteria such as Lachnospiraceae and Bacteroidetes and support microbial diversity and functional capacity [54,55].

Multivariate analysis revealed that dietary components of protein and vegetables differentially modulate faecal microbiota composition across cohorts, even in *Blastocystis*-colonised individuals. Some studies have shown that higher total protein intake (animals plus vegetables) correlates with higher microbial diversity and changes in bacterial profiles. Furthermore, evidence from human cohorts indicates that vegetable-rich diets promote greater bacterial richness and the presence of functional genera [56,57]. Furthermore, studies on dietary patterns have shown that plant-based diets are strongly associated with increased genera such as *Alistipes* [52] and *Sellinomas* [58]. In contrast, dietary styles characteristic of more restrictive environments favour distinct bacterial profiles [59]. The association of our proposed genera *Sellimonas* and *Murimonas* with healthier diets [60] and *Haemophilus* with more limited diets [61] reinforces the notion that *Blastocystis* may be shaped by host-specific ecological contexts. This supports the idea that its role should not be regarded as universally pathogenic. Consequently, its role should be analysed while considering nutritional quality and social and environmental determinants.

### 4.6. Strengths and Limitations

This study presents several notable strengths. First, it focuses on individuals colonised by *Blastocystis spp.*, allowing for a targeted analysis of dietary and ecological factors that shape the gut microbiota independent of colonisation status. By controlling for this key variable, we were able to examine microbiota structure within a relatively homogeneous colonisation profile, thereby reducing confounding effects. Second, we integrated detailed dietary quality assessment using the HEI-2020, enabling the exploration of specific components (e.g., protein and vegetable intake) and their multivariate associations with gut microbiota. Third, the inclusion of two contrasting populations—university students and institutionalised children/caregivers—provided an opportunity to examine the impact of diverse socioeconomic and dietary environments within a shared colonisation context. Lastly, our use of Hellinger-transformed genus-level data and Bray–Curtis dissimilarity in multivariate frameworks (PERMANOVA, db-RDA) ensured robust ecological interpretations.

However, several limitations must be acknowledged. The sample size was relatively small, particularly within each subgroup, limiting statistical power and the generalizability of results. Although our focus on *Blastocystis*-colonised individuals strengthens internal validity, it precludes direct comparisons with non-colonised controls. Furthermore, we did not perform subtyping of *Blastocystis*, which may be a relevant limitation given the reported variation in host responses and microbial associations for subtypes such as ST1, ST4, and ST7. Our cross-sectional design also limits causal inference, and while dietary data were collected using validated methods, reliance on self-reported intake may introduce recall bias. Finally, microbiota profiling was based on 16S rRNA sequencing at the genus level, which, while informative, lacks resolution on strain-specific or functional dynamics that could be captured through metagenomics.

Future studies should incorporate *Blastocystis* subtyping. While this study focused on individuals colonised by *Blastocystis*, we acknowledge that the absence of subtyping limits the ability to assess subtype-specific effects on the gut microbiota. Previous studies have shown that different *Blastocystis* subtypes may vary in their interactions with the host and microbial communities. Therefore, subtyping is essential for advancing our understanding of the ecological role of *Blastocystis* and for drawing more specific or causal inferences. It is also necessary expand to larger and more diverse cohorts and integrate metagenomic and metabolomic analyses to further disentangle the diet–microbiota–*Blastocystis* triad. Moreover, biological and contextual variables such as age, diet, and living conditions must also be considered when interpreting our findings. The differences in gut microbial diversity and structure between the FACSA and PAVILA cohorts are likely influenced by a combination of age, diet, and everyday living conditions. In the PAVILA cohort, the younger age of participants may partly explain their lower microbial diversity, as the gut microbiota continues to mature throughout childhood and adolescence. However, this represents only part of the picture. Institutionalised environment, where meals follow set menus and often depend on food donations, can limit dietary variety and reduce exposure to essential nutrients that support a healthy microbiome. These findings highlight the importance of considering age, diet quality, and social context as interconnected factors when interpreting gut microbiota profiles in individuals colonised by *Blastocystis*.

## 5. Conclusions

Our findings suggest that diet may play an important role in shaping gut microbiota composition among individuals colonised by *Blastocystis*, with differences potentially influenced by social and dietary environments. Although colonisation was present in both cohorts, the most pronounced differences in microbial diversity and structure appeared to reflect variation in diet quality, age, and living conditions. In particular, protein and vegetable intake were among the dietary components associated with microbial community patterns.

These findings contribute to the growing body of evidence suggesting that *Blastocystis* may act, under certain conditions, as an ecological marker of gut diversity rather than as a pathogenic agent. However, given the exploratory nature of this study and the small sample size, these associations should be interpreted with caution. Contextual factors such as diet, age, hygiene, and environment will likely influence colonisation and microbial dynamics in complex ways.

Future research should include *Blastocystis* subtyping and expand to larger, more diverse populations. Integrating metagenomic, metabolomic, and immunological data will be essential to better understand *Blastocystis’s* functional role and its interactions with the host and the microbiota.

## Figures and Tables

**Figure 1 microorganisms-13-01949-f001:**
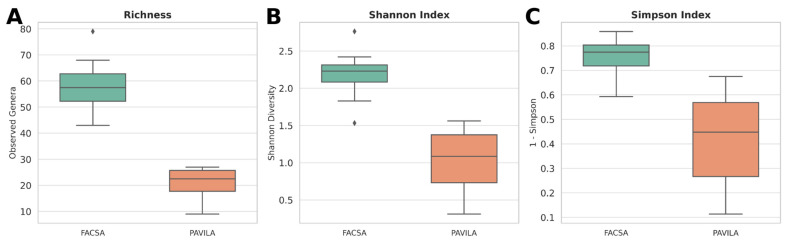
Alpha diversity comparison between cohorts colonised by *Blastocystis* spp. Boxplots show (**A**) richness (number of observed genera), (**B**) Shannon diversity index, and (**C**) Simpson diversity index (1–D) among individuals from the FACSA (university students) and PAVILA (institutionalised children) cohorts. Points represent individual samples; horizontal lines denote medians. FACSA individuals exhibited significantly higher richness and diversity across all indices compared to PAVILA (*p* < 0.05 for all tests, Wilcoxon rank-sum).

**Figure 2 microorganisms-13-01949-f002:**
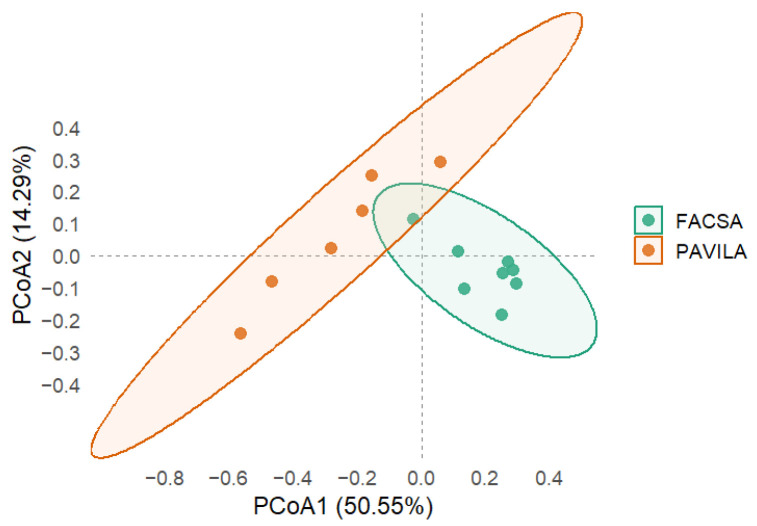
Principal Coordinates Analysis (PCoA) based on Bray–Curtis dissimilarities of genus-level microbiota profiles in FACSA and PAVILA cohorts. Each point represents an individual sample, and ellipses indicate 95% confidence intervals around group centroids. The first two PCoA axes explain 50.55% and 14.29% of the total variance, respectively. Group separation was statistically significant (PERMANOVA, F = 7.15, R^2^ = 0.374, *p* = 0.001).

**Figure 3 microorganisms-13-01949-f003:**
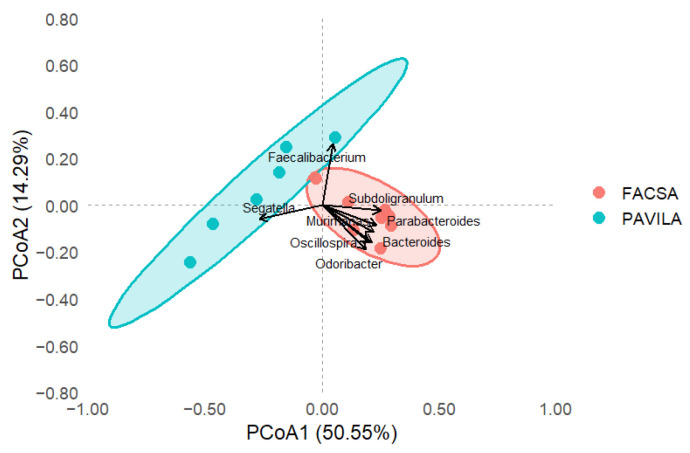
Principal coordinates analysis (PCoA) biplot showing the top 10 bacterial genes that are most aligned with Bray-Curtis dissimilarity axes. Arrows indicate the genus direction and magnitude of correlation. Samples are coloured by cohort (FACSA = orange, PAVILA = blue) and enclosed by 95% confidence ellipses.

**Figure 4 microorganisms-13-01949-f004:**
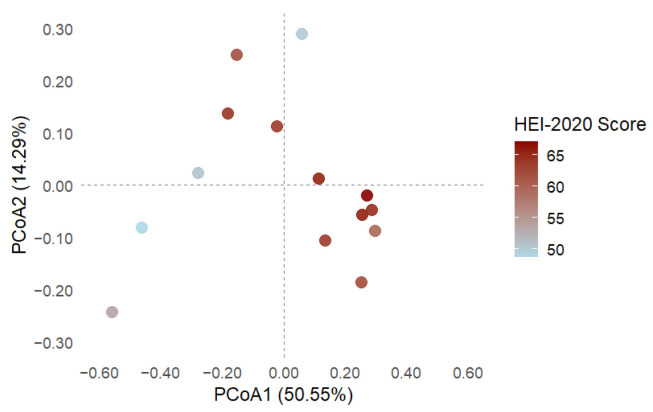
Principal coordinates analysis (PCoA) of gut microbiota profiles at the genus level based on Bray–Curtis dissimilarity among individuals colonised by *Blastocystis* from two cohorts. The HEI-2020 total diet quality score colours points. The first two PCoA axes explain 50.55% and 14.29% of the variance, respectively. A transparent ordination gradient is visible, with samples exhibiting higher dietary quality (red tones) clustering toward the right of the PCoA1 axis. In contrast, samples with lower HEI-2020 scores (represented by blue and purple tones) are positioned to the left. PERMANOVA confirmed a significant association between microbial composition and dietary quality (R^2^ = 0.24, *p* = 0.011), indicating that the host diet strongly influences the gut microbiota structure, even in the presence of *Blastocystis* colonisation.

**Figure 5 microorganisms-13-01949-f005:**
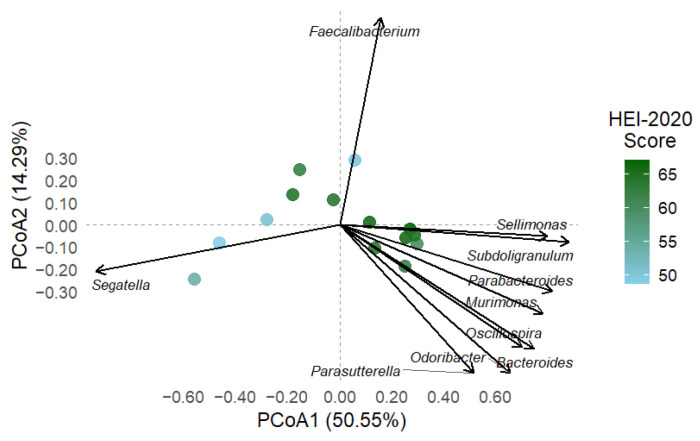
Biplot of the top 10 genera significantly associated with gut microbiota composition based on HEI-2020 diet quality scores (envfit, *p* < 0.1). The plot is based on Bray–Curtis dissimilarities and the first two PCoA axes, which explain 50.55% and 14.29% of the total variance, respectively. Arrows indicate the direction and strength of the association between each taxon and the compositional gradient. Genera such as *Sellimonas* and *Subdoligranulum* were aligned with higher HEI-2020 scores, while *Segatella* was associated with lower-quality diets.

**Figure 6 microorganisms-13-01949-f006:**
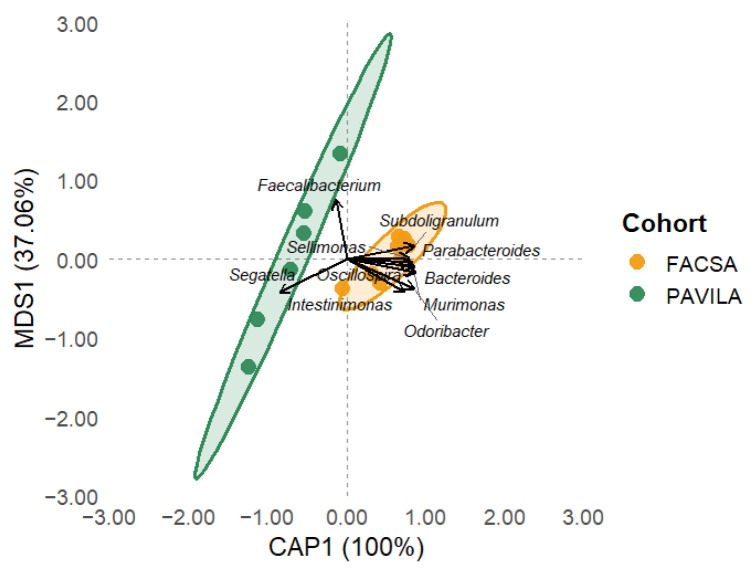
db-RDA constrained by Protein Foods Score. Arrows represent the 10 most correlated genera aligned with Bray-Curtis ordination. Ellipses represent 95% confidence intervals by cohort.

**Figure 7 microorganisms-13-01949-f007:**
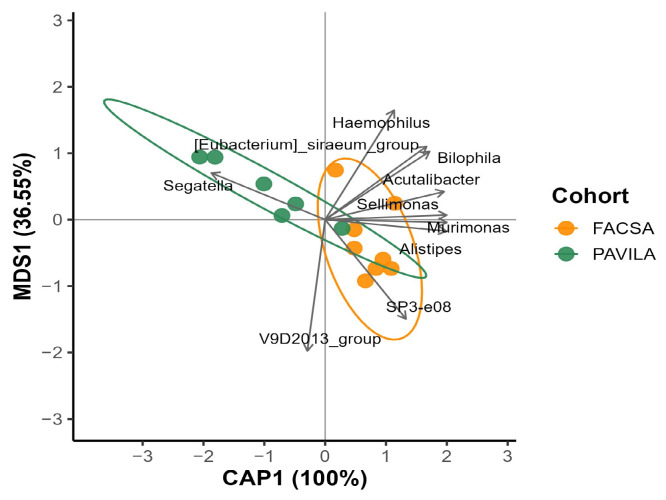
Distance-based redundancy analysis (db-RDA) of genus-level microbiota profiles in *Blastocystis*-colonised individuals, constrained by the vegetable intake component of the HEI-2020. Bray–Curtis dissimilarities were calculated after the Hellinger transformation. The first canonical axis (CAP1) explained 100% of the constrained variation. It separated the two cohorts, with FACSA (orange) projecting to the right, indicating higher vegetable intake, and PAVILA (green) to the left. Arrows represent the ten genera most strongly associated with the ordination axes (envfit, *p* < 0.1). Genera such as *Sellimonas*, *Acutalibacter*, *Murimonas*, and *Alistipes* were associated with higher vegetable intake, whereas *Segatella* was associated with lower scores in the PAVILA cohort.

**Figure 8 microorganisms-13-01949-f008:**
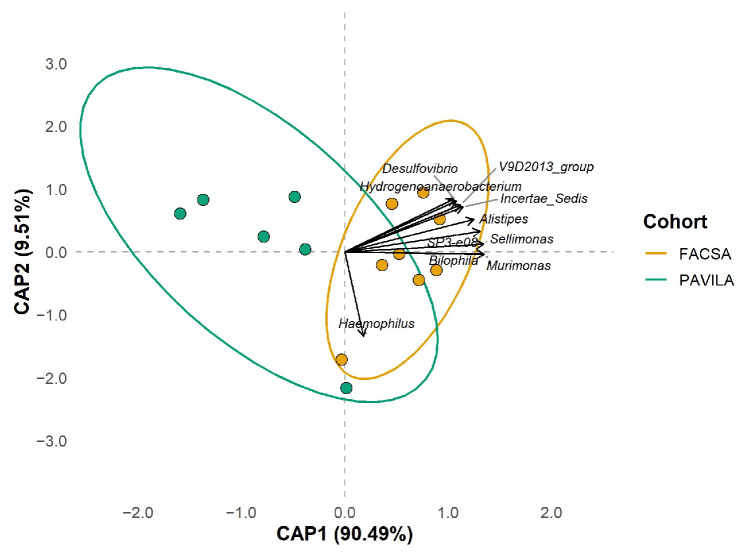
Distance-based redundancy analysis (db-RDA) using protein and vegetable dietary scores as explanatory variables. We performed the analysis using Bray–Curtis distances of Hellinger-transformed data. The points represent *Blastocystis*-colonised individuals from the FACSA (orange) and PAVILA (green) cohorts, and the ellipses indicate the 95% confidence intervals of each group. The ten taxa most associated with the canonical axes are included, projected by vectors (envfit, *p* < 0.1). The separation between cohorts suggests that the combined effect of these dietary components differentially modulates gut microbial composition depending on the host context.

**Table 3 microorganisms-13-01949-t003:** PERMANOVA results evaluating the association between each dietary component (HEI-2020) and gut microbiota composition.

Diet Component (HEI-2020)	R^2^	F	*p*-Value
Protein foods score	0.232	3.63	0.017
Total vegetable score	0.167	2.40	0.044
Total fruit score	0.132	1.82	0.117
Whole fruit score	0.135	1.88	0.117
Legume score	0.097	1.29	0.260
Refined grains score	0.092	1.21	0.279
Whole grain score	0.075	0.97	0.408
Saturated fats score	0.070	0.91	0.421
Sodium score	0.071	0.92	0.464
Seafood and plant protein score	0.064	0.82	0.482

Significance was assessed with PERMANOVA (vegan::adonis2, Bray–Curtis distance; 999 permutations). R^2^ = variance explained; F = pseudo-F statistic; *p* = permutation-based *p*-value **Pr(>F)** (right-tailed). Statistical significance was defined as *p* < 0.05 (α = 0.05).

**Table 4 microorganisms-13-01949-t004:** PERMANOVA Model Summary (Protein and Vegetables).

Term	Df	Sum of Squares	R^2^	F	*p*-Value
Model (Protein + Vegetables)	2	0.66195	0.31268	2.5021	0.025
Residual	11	1.45506	0.68732		
Total	13	2.11701	1.0		

Significance was assessed with PERMANOVA (vegan::adonis2, Bray–Curtis distance; 999 permutations). R^2^ = variance explained; F = pseudo-F statistic; *p* = permutation-based *p*-value **Pr(>F)** (right-tailed). Statistical significance was defined as *p* < 0.05 (α = 0.05).

## Data Availability

The data presented in this study are available in the article and Appendix A. Appendix A include anonymised dietary intake and microbiota metadata (n = 14), genus-level microbiota abundance, and R scripts used for statistical analysis. We provide all files to support transparency and reproducibility by the study’s ethical approval (CEI-FAMEN-36).

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
