# Peer review of "Diet Quality Modulates Gut Microbiota Structure in Blastocystis-Colonised Individuals from Two Distinct Cohorts with Contrasting Sociodemographic Profiles"

_microorganisms, 2025, doi:10.3390/microorganisms13081949_

Round 1

Reviewer 1 Report

Comments and Suggestions for Authors

This article evaluated the relationship between diet, gut microbiota, and Blastocystis colonisation in two different cohorts (university students (FACSA) and institutionalised children with their caregivers (PAVILA)). These findings highlight the role of diet in shaping the composition of gut microbiota. However, the relationship between Blastocystis colonization, gut microbiota, and diet has not been elucidated. As the authors note, ‘Although our focus on Blastocystis-colonised individuals strengthens internal validity, it precludes direct comparisons with non-colonised controls.’

The reviewers believe that the present study aims to highlight the modulation effects of dietary quality on gut microbiota structure in Blastocystis colonisation individuals. However, the relationship between Blastocystis colonisation, gut microbiome, and dietary quality does not appear to have been elucidated. The conclusion is generalisable, as previous literature has already demonstrated that dietary differences regulate the gut microbiome. (See references for details.)

Therefore, forcing this conclusion to be associated with Blastocystis colonisation seems overly contrived.

References

[1] Zmora, N., Suez, J. & Elinav, E. You are what you eat: diet, health and the gut microbiota. Nat Rev Gastroenterol Hepatol 16, 35–56 (2019). https://doi.org/10.1038/s41575-018-0061-2

[2] Conlon, M.A.; Bird, A.R. The Impact of Diet and Lifestyle on Gut Microbiota and Human Health. Nutrients 2015, 7, 17-44. https://doi.org/10.3390/nu7010017

  1. In Abstract. Please describe the reasons for selecting university students and institutionalised children with their caregivers as two cohorts in the sociodemographic environment. (Page 1, Line 31)
  2. In Abstract. The conclusion seems a little hasty. (Page 1, Line 32-34)
  3. In Introduction.The second paragraph mainly discusses the relationship between Blastocystis colonisation and the gut microbiota. However, this does not seem to be particularly relevant to the conclusions of the study.
  4. In Introduction. “Based on these observations, the present study aimed to investigate the relationship between Blastocystis colonisation and gut microbiota composition in individuals from two cohorts with different dietary patterns.” However, subsequent scheme designs did not highlight the significance of Blastocystis colonisation.
  5. In Materials and Methods.

-in Study Population and Design

-Please describe the age, gender, and other information of the volunteers.

- Please list the details. The ratio of children to their caregivers (n=37). In addition, the specific ratio of children to their caregivers colonized by Blastocystis.

-in Detection of Blastocystis

-Is there also a difference in the intensity of Blastocystis colonisation?

  1. 3.1. Dietary Quality and Blastocystis Status by Cohort

- “In the FACSA cohort, colonised participants exhibited higher total HEI-2020 scores, with a tendency toward greater intake of fruits and vegetables (Table 1). In contrast, in the PAVILA cohort, colonised children showed significantly lower vegetable intake compared to non-colonised peers.” What is the reason for this?

  1. 3.4 PCoA Biplot with Genus-Level Arrows

- “This pattern suggests that the microbiota of university students (FACSA), despite colonisation by Blastocystis, remains more diverse and enriched in genera typically linked to metabolic health and dietary fiber intake”. This conclusion seems to suggest that Blastocystis colonisation has little effect. Dietary factors have a greater influence. However, dietary factors are indeed universal.

-“the ecological impact of contrasting host environments such as institutionalisation and youth in the PAVILA cohort versus independent adulthood in the FACSA cohort.” What does this mean?

  1. 3.5. Diet–Microbiota Associations within Colonised Individuals

-This section does not explain the relationship between Blastocystis colonisation and the gut microbiota.

  1. In Discussion.

-Page 18, Line 681-686, this does not seem to be related to Blastocystis colonisation.

- Page 18, Line 694-695. Perhaps diet quality plays an even greater role?

  1. In Conclusions

-Page 22, Line 871-872. This conclusion is a universal law.

Author Response

Comment 1, Reviewer 1:
Please describe the reasons for selecting university students and institutionalised children with their caregivers as two cohorts in the sociodemographic environment (Page 1, Line 31).

Response to Comment 1 – Reviewer 1:
Thank you very much for your comment. In response, we have revised the Abstract to explicitly justify the selection of the two cohorts. We have now clarified that university students (FACSA) and institutionalised children with their caregivers (PAVILA) represent distinct dietary and sociodemographic contexts. This contrast enabled us to investigate how environmental, nutritional, and social factors might influence gut microbiota composition in individuals colonised by Blastocystis. This clarification is now included in the opening paragraph of the revised Abstract (Page 1, Line 15-19).

Comment2, Reviewer 1:
The conclusion seems a little hasty (Page 1, Line 32-34).

Response Comment 2, Reviewer 1:
We agree and have softened the conclusion in the Abstract. This now reads:

“Our results indicate that diet quality significantly influences gut microbiota composition in individuals colonised by Blastocystis, underscoring its potential as a target for nutritional interventions in vulnerable populations.”.

Introduction

Comment 3, Reviewer 1:
The second paragraph mainly discusses the relationship between Blastocystis colonisation and the gut microbiota. However, this does not seem to be particularly relevant to the conclusions of the study.

Response to Comment 3 – Reviewer 1:
We thank the reviewer for this comment. We have revised the Introduction to reduce the emphasis on Blastocystis, presenting it instead as a contextual biological factor. The primary focus is now clearly placed on how dietary quality modulates gut microbiota composition. The restructured Introduction reflects this revised emphasis more accurately.

Comment 4, Reviewer 1:

“Based on these observations, the present study aimed to investigate the relationship between Blastocystis colonisation and gut microbiota composition in individuals from two cohorts with different dietary patterns.” However, subsequent scheme designs did not highlight the significance of Blastocystis colonisation.

Response Comment 4, Reviewer 1:

Thank you for your comment. We have revised the objective to clarify that Blastocystis colonisation was used as a contextual framework for exploring diet microbiota interactions, rather than as a primary outcome or central analytical variable.

Materials and Methods

Comment 5, Reviewer 1:

Please describe the age, gender, and other information of the volunteers.

- Please list the details. The ratio of children to their caregivers (n=37). In addition, the specific ratio of children to their caregivers colonized by Blastocystis.

Response Comment 5, Reviewer 1:

We thank the reviewer for these comments. The requested data has been added in the manuscript (lines 107-116).

Comment 6, Reviewer 1:
Is there also a difference in the intensity of Blastocystis colonisation?

Response Comment 6, Reviewer 1:

We thank the reviewer for this observation. As noted in the revised manuscript (lines 196-199), the assay was conducted as a qualitative test to detect the presence or absence of Blastocystis. No evaluation of colonisation intensity was performed, as quantification of parasite load was beyond the scope of this study.

Results

Comment 7, Reviewer 1:

  1. 3.1. Dietary Quality and Blastocystis Status by Cohort

“In the FACSA cohort, colonised participants exhibited higher total HEI-2020 scores [...] In contrast, in the PAVILA cohort, colonised children showed significantly lower vegetable intake compared to non-colonised peers.” What is the reason for this?

Response to Comment 7 – Reviewer 1:
Thank you for your thoughtful comment. We agree that the contrasting dietary patterns observed in relation to Blastocystis colonisation required further clarification. In response, we have added a paragraph to the Discussion section (lines 739-755) offering a contextual interpretation. Specifically, we hypothesise that in the PAVILA cohort, institutional food planning combined with weekend return to unsupervised home environments may reduce access to fresh vegetables, potentially influencing colonisation dynamics. In contrast, FACSA students showed higher HEI-2020 scores among colonised individuals, in line with previous evidence suggesting that fibre-rich diets may favour colonisation. We further discuss the influence of hygiene, immune development, and environmental exposure, highlighting that the relationship between diet and Blastocystis is likely context-dependent and modulated by broader sociodemographic factors. These interpretations are supported by relevant epidemiological studies.

Comment 8, Reviewer 1:

  1. Comment 8 Reviewer 1:

3.4 PCoA Biplot with Genus-Level Arrows. Is now 3.3

- “This pattern suggests that the microbiota of university students (FACSA), despite colonisation by Blastocystis, remains more diverse and enriched in genera typically linked to metabolic health and dietary fiber intake”. This conclusion seems to suggest that Blastocystis colonisation has little effect. Dietary factors have a greater influence. However, dietary factors are indeed universal.

Response Comment 8 Reviewer 1: We appreciate this insight. The sentence has been revised to clarify that dietary and environmental factors likely override any minor effect that Blastocystis colonisation might have on microbial diversity. We now frame Blastocystis more as an ecological marker rather than a causal agent, in line with current literature (lines 370-372).

Comment 9, Reviewer 1:

“the ecological impact of contrasting host environments...” What does this mean?

Response Comment 9, Reviewer 1:

We have revised this sentence to be more explicit. It now reads: “This reflects the distinct living conditions, levels of autonomy, and dietary environments between university students and institutionalised children, which likely modulate microbiota structure.”

Moreover, when referring to the “ecological impact” of host environments, we refer to the combined influence of factors such as institutionalisation, age, dietary control, and environmental exposure, which jointly shape the gut microbiota landscape in each cohort (lines 374-380).

Comment 10, Reviewer 1:

  1. 3.5. Diet–Microbiota Associations within Colonised Individuals

-This section does not explain the relationship between Blastocystis colonisation and the gut microbiota.

Response Comment 10, Reviewer 1:

This section focuses on diet–microbiota associations within colonised individuals. We now explicitly state this in the Results section and added a clarifying sentence to prevent misinterpretation (lines 703-709).

Discussion and Conclusions

Comment 11, Reviewer 1:
Page 18, Line 681-686, this does not seem to be related to Blastocystis colonisation.

Response Comment 11, Reviewer 1:
We agree with the reviewer. This sentence has been removed to maintain coherence with the main hypothesis (lines 732-738).

Comment 12, Reviewer 1:
Page 18, Line 694-695. Perhaps diet quality plays an even greater role?

Response Comment 12, Reviewer 1:
Yes, we agree with the reviewer and have now revised the sentence to give more weight to dietary quality as a modulator of microbiota structure, especially in colonised individuals. Lines 774-779.

Comment 13, Reviewer 1:
Page 22, Line 871-872. This conclusion is a universal law.

Response Comment 13, Reviewer 1:

We thank the reviewer for this observation. In response to your comment and to those of other reviewers, we have substantially revised the final conclusion to avoid overgeneralisation and better reflect the scope of our findings. The updated conclusion can be found in lines 974-990 of the revised manuscript.

Reviewer 2 Report

Comments and Suggestions for Authors

Comments for authors

While the study addresses a relevant and emerging topic in microbiota research and utilizes appropriate methodology, several issues need clarification before publication.

Decision: Major Revision

Abstract

Needs greater clarity in population characteristics—e.g., mention that only Blastocystis-colonised subjects were analyzed in sequencing. Clearly state sample size of microbiota-analyzed participants (n=14).

Introduction

Streamline the discussion on conflicting roles of Blastocystis (avoid saying the same point in multiple ways).

Clarify what is already known and what the study aims to newly address.

Materials and Methods

Small sample size for microbiota analysis (n=14) not emphasized. Please justify why only 14 samples were analyzed.

Results

Emphasize the exploratory nature of analyses due to small n

Discussion

Discuss more explicitly that Blastocystis subtyping is essential for drawing causal conclusions.

Conclusion

Reframe “acts more as an ecological indicator than a pathogen” to reflect the exploratory nature of the study.

Minor comments

Some typo mistakes present, for example, in line 647, "We perform The analysis using…." Should be "we performed the analysis using ……”

Some abbreviations (e.g., SCFA) should be defined earlier.

Author Response

Comment 1, Reviewer 2: Needs greater clarity in population characteristics—e.g., mention that only Blastocystis-colonised subjects were analyzed in sequencing. Clearly state sample size of microbiota-analyzed participants (n=14).

Response to Comment 1 – Reviewer 2:
We thank the reviewer for your suggestion. We have revised the Abstract to clearly state that only Blastocystis-colonised individuals were included in the microbiota analysis and to indicate the sample size (n=14). These clarifications have been added in lines 17–19 of the revised manuscript.

Introduction

Comment 2 – Reviewer 2:
Streamline the discussion on conflicting roles of Blastocystis (avoid saying the same point in multiple ways). Clarify what is already known and what the study aims to newly address.

Response to Comment 2 – Reviewer 2:
We thank the reviewer for the insightful observation. In response, we have revised the Introduction to streamline the presentation of the dual role of Blastocystis, removing redundant statements. In addition, we have also clarified the current state of knowledge and highlighted the novel contributions of this study. These revisions are reflected in the updated Introduction, particularly in lines (93-101).

Materials and Methods

Comment 3 – Reviewer 2:
Small sample size for microbiota analysis (n=14) not emphasized. Please justify why only 14 samples were analyzed.

Response to Comment 3 – Reviewer 2:
We thank the reviewer for this comment. We have now described in the Methods section (lines 120–125) that only 14 samples were analysed due to financial and logistical constraints related to sequencing. We also emphasised the exploratory nature of the study and noted that this small sample size is appropriate for hypothesis generation rather than for drawing definitive conclusions.

Results

Comment 4 – Reviewer 2:
Emphasize the exploratory nature of analyses due to small n.

Response comment Reviewer 2

We thank the reviewer for this observation. We have incorporated a sentence at the beginning of the Results section (lines 244-245) to highlight the exploratory nature of the analyses, given the small sample size. We believe that this provides appropriate context for the interpretation of our findings.

Discussion

Comment 5 – Reviewer 2:
Discuss more explicitly that Blastocystis subtyping is essential for drawing causal conclusions.

Response to Comment 5 – Reviewer 2:
We thank the reviewer for this insightful comment. We have expanded the Discussion section to explicitly acknowledge the importance of Blastocystis subtyping for interpreting microbiota associations and for drawing any causal conclusions. We have also highlighted that the absence of subtyping is a limitation of our study. These additions are included in lines 958-964 of the revised manuscript.

Conclusion

Comment 6 – Reviewer 2:
Reframe “acts more as an ecological indicator than a pathogen” to reflect the exploratory nature of the study.

Response to Comment 6 – Reviewer 2:
We agree with this reviewer’s suggestion. The conclusion has been reworded to reflect the exploratory scope of the study and to avoid definitive statements. The new phrasing appears in lines 981-985 of the revised manuscript.

Minor Comments

Comment 7 – Reviewer 2:
Some typo mistakes present, for example, in line 647, "We perform The analysis using…” should be “we performed the analysis using ……”

Response comment 7 – reviewer 2:
We agree with the reviewer. The grammatical error in line 697 (along with other minor typographical mistakes) have been corrected throughout the manuscript.

Comment 8 – Reviewer 2:
Some abbreviations (e.g., SCFA) should be defined earlier.

Response to Comment 8 – Reviewer 2:
We agree with the reviewer and have carefully reviewed the manuscript and confirmed that all abbreviations, including SCFA (short-chain fatty acid), are defined at their first mention. Specifically, SCFA is introduced in line 75 as “short-chain fatty acid (SCFA).” We also ensured that other abbreviations such as PERMANOVA, QIIME2, ASV, and PCR are now properly defined upon first use.

Reviewer 3 Report

Comments and Suggestions for Authors

This study examines diet, gut microbiota, and Blastocystis in two groups (students and institutionalized children/caregivers). Better diet quality (higher protein/vegetable intake) was linked to beneficial gut bacteria, while poorer diets correlated with less favorable microbes. Despite all having Blastocystis, diet and living conditions influenced microbiota, suggesting dietary interventions could improve gut health in vulnerable populations. However, the manuscript is organized poorly, especially the Figures should be redrawn, and the discussion of this manuscript is also weak; a strong comparison should be made with previous studies, and citations should the obtained where needed. The specific comments and suggestions are as follows.

  1. "?????" The comma should be deleted from the Title
  2. Line 23" p < 0.01", this should be like this " p < 0.01" followed throughout the article.
  3. Lines 43-44: "...........system [1,2] that greatly contributes to metabolic homeostasis [3]....................... the general wellbeing [5]" Language should be changed for this.
  4. Lines 56-57: " In addition, some studies .......................obesity, and depression [19]." More references should be cited in this.
  5. Lines 89-96: In children living in institutional settings..................... economic backgrounds" reference should be cited in this.
  6. Headings 3.2 and 3.3 should be combined under one heading.
  7. Figure 2: should be redrawn by putting a line at x- and the y-axis. Similar should be followed for the other Figures.
  8. Figure 4 should be modified using a line on the x-axis and y-axis.
  9. Lines 666-669: "In this study,....................students (FACSA) and institutional." This should be deleted.
  10. Lines 673-687: "Diet quality, as quantified by HEI-2020 scores, differed significantly between..............gut microbial community structure [39–41]." This paragraph showed the literature review, which should make a comparison of this study with previous reports.
  11. Line 692-696: "In our analysis focusing exclusively...................evenness compared to PAVILA participants." This is the objective of this study, not the results.
  12. Line 871- 873: "Our findings highlight the critical............... environments." The language for this should be revised.

Author Response

Response to Reviewer 3

Comment 1 – Reviewer 3: The comma should be deleted from the Title

Response Comment 1-Reviewer 3:

We thank the reviewer for spotting this. The comma has been removed from the title.

Comment 2 – Reviewer 3: Line 23: 'p < 0.01', this should be like this 'p < 0.01' followed throughout the article.

Response Comment 2 – Reviewer 3:

We thank the reviewer for the suggestion. We have reviewed the entire manuscript and formatted all *p*-values in italics (e.g., *p* < 0.01), in accordance with journal style guidelines. Consistency has been applied throughout the text.

Comment 3 Reviewer 3:  Lines 43-44: "...........system [1,2] that greatly contributes to metabolic homeostasis [3]....................... the general wellbeing [5]" Language should be changed for this.

Response to Comment 3 – Reviewer 3:
We thank the reviewer for pointing this out. The sentence has been rephrased for improved clarity and accuracy. The revised version can be found in lines 42–46 of the revised manuscript.

Comment 4 Reviewer 3: Lines 56-57: " In addition, some studies .......................obesity, and depression [19]." More references should be cited in this.

 Response to Comment 4 – Reviewer 3:
We thank the reviewer for this suggestion. As part of the Introduction’s restructuring in response to comments from other reviewers, the paragraph in question was removed. Consequently, the sentence referencing associations with obesity and depression is no longer included in the revised version.

Comment 5 – Reviewer 3: Lines 89-96: In children living in institutional settings..................... economic backgrounds" reference should be cited in this.

Response to Comment 5 – Reviewer 3:
We thank reviewer’s suggestion, and we have added two supporting references to substantiate the statement regarding children living in institutional settings. Due to the restructuring of the Introduction, the citations now appear in lines 81-86 of the revised manuscript.

Comment 6 – Reviewer 3: Headings 3.2 and 3.3 should be combined under one heading.

Response to Comment 6 – Reviewer 3:
Thank you for your suggestion. We have combined the sections on alpha and beta diversity under a single heading entitled: “3.2. Alpha and Beta Diversity Analyses.” This modification improves the coherence of the Results section and avoids redundancy.

Comment 7- Reviewer 3: Figure 2: should be redrawn by putting a line at x- and the y-axis. Similar should be followed for the other Figures.

Response to Comment 7 – Reviewer 3:
We agree with the reviewer’s suggestion. Figure 2 has been redrawn to include lines on both the x- and y-axes to improve visual clarity. The same adjustment has been applied to the other figures as needed to ensure consistency throughout the manuscript.

Comment 8 – Reviewer 3: Figure 4 should be modified using a line on the x-axis and y-axis.

Response to Comment 8 – Reviewer 3:
Thank you for your observation. Figure 4 has been updated to include lines on both the x- and y-axes, in line with the adjustments made to other figures for consistency and improved visual presentation.

Comment 9 – Reviewer 3: Lines 666-669: "In this study,....................students (FACSA) and institutional." This should be deleted.

Response to Comment 9 – Reviewer 3:
We thank the reviewer’s suggestion. We understand the concern regarding redundancy and agree that the original sentence was not essential in its initial form. Rather than removing it entirely, we have rephrased the sentence to enhance its clarity and relevance within the context of the Discussion. The revised version now contributes more directly to the paragraph’s main argument and can be found in lines 715–720 of the updated manuscript.

Comment 10 – Reviewer 3: Lines 673-687: "Diet quality, as quantified by HEI-2020 scores, differed significantly between..............gut microbial community structure [39–41]." This paragraph showed the literature review, which should make a comparison of this study with previous reports.

Response to Comment 10 – Reviewer 3:
We thank the reviewer for this important observation. In response, we have substantially revised the paragraph to include direct comparisons between our findings and previous literature. We now contrast the differential patterns of diet quality and Blastocystis colonisation across cohorts with existing studies, and we discuss the influence of socioeconomic and environmental factors. The revised paragraph includes new references (Conlon & Bird, 2015; Deehan et al., 2020; Um et al., 2023; Piperni et al., 2024; Salazar-Sánchez et al., 2021; Marangi et al., 2023) and can be found in lines 739–755 of the updated manuscript.

Comment 11 – Reviewer 3: Line 692-696: "In our analysis focusing exclusively...................evenness compared to PAVILA participants." This is the objective of this study, not the results.

Response to Comment 11 – Reviewer 3:
Thank you for your observation. We agree that the original phrasing resembled an objective rather than a result. We have revised the sentence to present it as a finding from our analysis. The updated version now reads:
“When comparing gut microbiota diversity among Blastocystis-colonised individuals, we observed that participants from the FACSA cohort exhibited significantly higher microbial richness and evenness than those from the PAVILA cohort, suggesting a strong influence of host-related factors.”
This revision appears in lines 764–767 of the manuscript.

Comment 12 – Reviewer 3: Line 871- 873: "Our findings highlight the critical............... environments." The language for this should be revised.

Response: Thank you for this suggestion. The conclusion has been revised for clarity and to reflect the exploratory nature of our findings. The updated version is found in lines 974-980 of the manuscript.

Reviewer 4 Report

Comments and Suggestions for Authors

The paper presents a well-designed study that explores how diet quality modulates gut microbiota diversity and structure in individuals colonized by Blastocystis. For the experiments, the authors compared two socioeconomic and demographically distinct groups (university students vs. institutionalized children/caregivers). The work integrates microbiome sequencing (16S rRNA), dietary data, and statistical ecology to address a gap in understanding the diet–microbiome–protozoan interface.
However, several areas could benefit from refinement. Below are detailed suggestions.
Abstract could mention sample size in the final sentence to temper expectations. Clarify whether all participants were colonized by Blastocystis or only the microbiome subcohort (14 subjects).
The Introduction is slightly long and repetitive in some areas (e.g., roles of Blastocystis, diet–microbiome links).
I would recommend using a stronger focus for the research question.
The Discussion of this manuscript is scientifically well-structured, conceptually robust, and clear. The authors effectively integrate ecological, microbiological, and nutritional evidence to contextualize their findings. The comparative analysis between two distinct cohorts, under the unifying condition of Blastocystis colonisation, adds depth and novelty to the interpretation.
While the authors discuss the influence of age on alpha and beta diversity, the interplay with diet is confusing. Are differences in microbial composition due to age-related maturation, diet, institutional context, or a combination? I would suggest adding a brief comparative reflection or schematic summary to distinguish these variables.
Finally, the manuscript is rich, original, and methodologically sound, and in the scope of the journal. I recommend publication after these minor revisions.

Author Response

Comment 1 Reviewer 4: Abstract could mention sample size in the final sentence to temper expectations. Clarify whether all participants were colonized by Blastocystis or only the microbiome subcohort (14 subjects).

Response Comment 1 Reviewer 4:

We thank the reviewer for this important observation. In response, we have revised the Abstract to clarify that only 14 individuals colonised by Blastocystis were included in the microbiota sequencing analysis. This clarification has been added to the second paragraph of the Abstract, immediately following the description of the two cohorts, to provide appropriate context regarding the analysed subcohort and its sample size.

Comment 2 Reviewer 4: The Introduction is slightly long and repetitive in some areas (e.g., roles of Blastocystis, diet–microbiome links). I would recommend using a stronger focus for the research question.

Response Comment 2 Reviewer 4:
We agree with this recommendation and have streamlined the Introduction to remove redundant elements, particularly regarding the dual role of Blastocystis and general statements about diet–microbiota associations. We have also revised the final paragraph of the Introduction to present a more focused research aim. These changes appear in lines 42-101 of the revised manuscript.

Comment 3 Reviewer 4: While the authors discuss the influence of age on alpha and beta diversity, the interplay with diet is confusing. Are differences in microbial composition due to age-related maturation, diet, institutional context, or a combination? I would suggest adding a brief comparative reflection or schematic summary to distinguish these variables.

Response Comment 3 Reviewer 4:
We than the reviewer for this insightful comment. We have revised the Discussion section to clarify the interaction between age, diet, and institutional context as co-occurring factors potentially shaping microbial composition. We acknowledge that our design does not allow these variables to be fully disentangled, but we now include a clearer comparative reflection to distinguish their likely contributions. In addition, we have added a supplementary figure (Supplementary Figure S1) that visually summarises the key contrasts between the two cohorts and the factors influencing gut microbiota diversity and structure (lines 961-971).

Round 2

Reviewer 1 Report

Comments and Suggestions for Authors

Revised according to the reviewers' comments.
However, the scientific hypothesis of the article seems to be universal. Please emphasise the significance of Blastocystis colonisation.

Round 2

Comment Reviewer 1: Revised according to the reviewers' comments.
However, the scientific hypothesis of the article seems to be universal. Please emphasise the significance of Blastocystis colonisation.

Response Comment Reviewer 1: We thank the Reviewer for this insightful observation. The role of Blastocystis colonisation should be more explicitly emphasised in our hypothesis. In the original version, this concept was implicit in the final paragraph of the Introduction, where we highlighted that Blastocystis colonisation may reflect environmental and nutritional exposures and that the role of diet in modulating this relationship is poorly understood.

We have revised the final paragraph of the Introduction to explicitly state our hypothesis: that the impact of Blastocystis on gut microbiota composition and diversity may vary according to diet quality and sociodemographic environment. This addition strengthens the focus on Blastocystis colonisation while remaining consistent with our original study aim.

The revised paragraph in the Introduction now reads:

In this context, Blastocystis colonisation may reflect environmental and nutritional exposures rather than pathogenicity. While the relationship between Blastocystis and the microbiota has been partially described, the role of dietary quality in modulating this interaction remains poorly understood. We hypothesised that the impact of Blastocystis on gut microbiota composition and diversity would vary according to diet quality and sociodemographic environment. Therefore, this study aimed to examine whether gut microbiota structure differs according to diet quality in Blastocystis-colonised individuals from two cohorts with contrasting dietary patterns and sociodemographic contexts: university students (FACSA) and institutionalised children with their caregivers (PAVILA). Lines 93-103

Reviewer 2 Report

Comments and Suggestions for Authors

The authors responded well to all comments and the manuscript greatly improved and become suitable for publication. 

Author Response

We sincerely thank the reviewer for the positive feedback and for recognising the improvements made to the manuscript. We genuinely appreciate time, effort, and constructive comments, which have strengthened the final version of our work.

Reviewer 3 Report

Comments and Suggestions for Authors

The authors carefully incorporate all the comments, and the article can be accepted for publication. 

Author Response

We sincerely thank the reviewer for the positive assessment and for acknowledging the revisions made. We are grateful for constructive feedback, which has helped improve our manuscript's quality and clarity.